# Whole-Exome Sequencing to Identify Potential Genetic Risk in Substance Use Disorders: A Pilot Feasibility Study

**DOI:** 10.3390/jcm10132810

**Published:** 2021-06-25

**Authors:** P. V. AshaRani, Syidda Amron, Noor Azizah Bte Zainuldin, Sumanty Tohari, Alvin Y. J. Ng, Guo Song, Byrappa Venkatesh, Ajay S. Mathuru

**Affiliations:** 1Research Division, Institute of Mental Health, Singapore 539747, Singapore; Asharani_PEZHUMMOOTTIL_VASUDEVAN_N@imh.com.sg; 2National Addictions Management Service, Institute of Mental Health, Singapore 539747, Singapore; syidda_amron@imh.com.sg (S.A.); na_zainuldin@imh.com.sg (N.A.B.Z.); song_guo@imh.com.sg (G.S.); 3Institute of Molecular and Cell Biology, Singapore 138673, Singapore; Sumanty_Tohari@ibn.a-star.edu.sg (S.T.); alvin_yj_NG@nuhs.edu.sg (A.Y.J.N.); 4Department of Pediatrics, Yong Loo Lin School of Medicine, National University of Singapore, Singapore 119228, Singapore; 5Yale-NUS College, Singapore 138610, Singapore; 6Department of Physiology, Yong Loo Lin School of Medicine, National University of Singapore, Singapore 117593, Singapore

**Keywords:** alcohol-dependence, substance use disorders, whole-exome sequencing, cohort pilot study, family trios

## Abstract

Genetics intersects with environmental, cultural, and social factors in the development of addictive disorders. This study reports the feasibility of whole-exome sequencing of trios (subject and two family members) to discover potential genetic variants in the development of substance use disorders (SUD). Family trios were recruited from the National Addictions Management Service in Singapore during the 2016–2018 period. Recruited subjects had severe alcohol use disorder (AUD) or opioid use disorder (OUD), with nicotine dependence (ND) and a family history of addictive disorders. Demographic characteristics and severity of addiction were captured. Whole-exome sequencing (WES) and analysis were performed on salivary samples collected from the trios. WES revealed variants in several genes in each individual and disruptive protein mutations in most. Variants were identified in genes previously associated with SUDs, such as Pleckstrin homology domain-containing family M member 3 (PLEKHM3), coiled-coil serine-rich protein 1 (CCSER1), LIM and calponin homology domains-containing protein 1 (LIMCH1), dynein axonemal heavy chain 8 (DNAH8), and the taste receptor type 2 member 38 (TAS2R38) involved in the perception of bitterness. The feasibility study suggests that subjects with a severe addiction profile, polysubstance use, and family history of addiction may often harbor gene variants that may predispose them to SUDs. This study could serve as a model for future precision medicine-based personalized interventional strategies for behavioral addictions and SUDs and for the discovery of potentially pathogenic genetic variants.

## 1. Introduction

Substance abuse is a leading cause of mortality and morbidity. The World Drug Report 2019 shows that around 5.5% of the world’s population aged 15–64 years used drugs in the preceding 12 months [1]. When it comes to the most serious impact of substance abuse, over 47,000 deaths were reported in 2017 due to opioid overdose in the United States alone, a 13% increase compared to the previous year [1]. Alcohol abuse accounted for ~3 million deaths in 2016, with over 2.7 billion estimated users aged 15 or above [2]. Approximately 1.1 billion people across the globe are current smokers, and smoking is estimated to kill 8 million people every year [3]. National-level, cross-sectional studies in Singapore have reported smoking prevalence to 16.1% of the population, with ~3.3% showing a dependence on nicotine [4]. Approximately 2500 deaths/year among smokers and 250 deaths among non-smokers are attributable to tobacco use [5]. A population-level study in Singapore showed that the lifetime prevalence of alcohol use and dependence were 4.1% and 0.5%, respectively [6].

Additionally, epidemiological evidence links smoking to psychiatric conditions and illicit substance use [7]. Almost three-quarters of illicit substance users reported current smoking with higher odds of smoking among polysubstance users [8]. In general, the risk of death is higher among polysubstance users compared to the general population accounting for both somatic diseases (58% vs. 28%, respectively) and overdose-related deaths (33% vs. 9%, respectively) [9]. A retrospective analysis of unnatural death among treatment-seeking substance abusers in Singapore showed a large proportion of suicides among subjects abusing alcohol and opiates [10]. Apart from the pervasive effects on an individual’s life, health, and health care utilization, substance misuse also impacts society and the economy [11]. Thus there is a pressing need for an in-depth understanding of the factors that contribute to the development of addiction and identify counter-strategies.

Among the affected, substance use starts at an early age and gradually develops into substance use disorder (SUD). Age of onset depends on the substance of abuse and several other factors such as availability, cultural precepts, and genetics. Discordant twin studies, for example, show that familial influences determine the variations in the age of onset for various substances [12]. The early age of onset of one substance, however, is causally linked to the initiation of a second substance and is a risk factor for SUD and the psychological consequences. Bierut et al., in their pioneering work, demonstrated that siblings of alcohol-dependent subjects have elevated rates of development of alcohol, cocaine, and marijuana dependence compared to siblings of controls [13]. A lifetime diagnosis of alcohol dependence (AD) was observed in 49.3–50.1% of brothers and 22.4–25% of sisters showing a strong familial aggregation. Twin studies of monozygotic or dizygotic twins over the years also suggest a heritability of 50–70% for developing dependence [14]. For example, a Swedish national-level study on AUD among twins collating data from medical pharmacy and criminal registries showed a heritability of 57% among males and 22% among females, respectively [15]. Finally, meta-analyses also showed substance-dependent heritability that ranges from 33–71% for ND, 48–66% for AD, 42–79% for cocaine [16], and 23–54% for opioids [17]. Given this evidence on familial segregation, numerous studies have been conducted since 2005 to identify putative candidate genes in addictive disorders [14].

Past genome-wide association studies (GWAS) have revealed at least 8 genetic loci for AUD, 11 for ND, and 2 for illicit drugs that showed a significant association with drug dependence across populations [14]. Although past GWASs have uncovered a number of single nucleotide polymorphisms (SNPs), many are in the noncoding regions and may not give an insight into the biological mechanisms affected in the context of the SUD. Although a large number of putative candidates from genomic studies exist, only a few genes, such as the alcohol dehydrogenase gene ALDH2 and ADH1B in AUD, two nicotinic acetylcholine receptor subunit genes CHRNA5, CHRNB4 in ND, and OPRM1 in opioid use disorder (OUD) have been examined extensively with respect to the potential neural circuits impacted using animal models [18] and the molecular mechanisms affected. Moreover, the variants discovered in GWAS at present are proposed to explain less than 10% of the dependence, necessitating further work and exploring other strategies [19]. One additional route is whole-exome sequencing or WES, typically of trios of a family (subject + parents/siblings). WES has been employed successfully to explore the medical genetics of a number of disorders using such an approach [20] and is considered a promising route to understand the genetic vulnerability related to SUD [19,21].

Previous studies have also shown that it is possible to successfully apply WES to identify the pathogenic variants in multi-genic complex disorders such as mental illness and hypertension through the deep sequencing of a few individuals or family trios [20]. However, it has not yet been applied to study genetic factors associated with SUD in specific populations. The feasibility of applying such a strategy in the Singapore context, specifically for the study of the development of dependence, also lacks precedence. In particular, how the sociological, familial, and psychological challenges that subjects with SUD face in the local context impacts the implementation of such a strategy is unclear [22]. Our study explored the feasibility of conducting WES in the treatment-seeking population at the National Addictions Management Service (NAMS) Clinic of Singapore with the intent of identifying potential pathogenic variants, candidate genes, and the challenges in conducting such an exercise when constructing SUD treatments. This feasibility analysis study suggests that recruitment can be challenging, but it identifies steps that can be taken to improve participation in future large-scale studies. It also informs on the likelihood of finding potential pathogenic variants by WES of family trios.

## 2. Materials and Methods

### 2.1. Participants

This cross-sectional study recruited treatment-seeking individuals and two family members from the NAMS and was conducted in the period of December 2016 to October 2018. Five family trios were recruited from the outpatient clinics or inpatient wards during the study period. The patient was asked to identify two of his family members who would be willing to take part in the study. The selected family members were approached for the study. Those family members who were willing were recruited to the study regardless of their substance use status. The study followed the protocol approved by the Domain Specific Review Board (DSRB Ref: 2016/01111). Written informed consent was also taken from all the participants.

### 2.2. Eligibility Criteria

The study recruited participants above the age of 21 years. Subjects were enrolled in the study if they had a *Diagnostic and Statistical Manual of Mental Disorders, 5th Edition (DSM-5)* diagnosis of AUD or OUD together with ND. In order to increase the chances of finding a polymorphism, subjects with higher severity of addictions were recruited. As such, subjects with a DSM-5 score of 8 or above were enrolled in the study. Other eligibility criteria included a self-reported family history of any type of addiction, willingness to enroll in the study with two immediate, genetically related family members (specifically, parents, grandparents, children, or siblings whose genetic relationship could be verified). Subjects who could not read English or had a diagnosis of bloodborne diseases (Hepatitis B, Acquired ImmunoDeficiency Syndrome, etc.) were excluded from the study.

### 2.3. Data Collection

The data collection forms captured basic socio-demographic information that included age, gender, education, ethnicity, nationality, and marital status. Other questionnaires used in the study are described below.

### 2.4. Substance Use Data Collection

#### 2.4.1. Alcohol Use Disorder and Opioid Use Disorder

DSM-5 was used to screen potential participants. Those who scored above 8 were included in the study. DSM-5 was administered to all five subjects who were patients registered with the addiction clinic and not their family members.

#### 2.4.2. Nicotine Dependence: Fagerstrom Test for ND

This 6-item questionnaire measures the quantity, compulsion to use, and physical dependence on nicotine. The items will add up to a score between 0 and 10. A score of 2 or less is considered low dependency; 3–4 is low-moderate, 5–7 is moderate dependency, and 8 or above is high dependency. The questionnaire was administered to all the participants.

#### 2.4.3. The Severity of Addiction

The Addiction Severity Index-Lite [23] (ASI lite) measures the severity of addictions in 6 domains: drug and alcohol use, employment, legal, medical, family/social, and psychiatric. A composite score is calculated for individual domains. The scores range from 0 to 1. A score of 0 indicates ‘no problems’, and 1 indicates ‘higher severity of problems’. Higher scores for all the individual domains indicate severe problems, except for the employment domain, where a higher score shows the strength of the employment. The scale was administered to all the participants.

#### 2.4.4. Alcohol Use Disorder Identification Test (AUDIT)

AUDIT is a 10-item self-reported measure to capture the alcohol use patterns and problems. The scores range from 0 to 40. A score of 8 or more is indicative of hazardous drinking, and a score of 20 and above indicates high risk of alcohol dependence that requires further clinical/diagnostic evaluation. All the participants answered this questionnaire.

#### 2.4.5. Collection of Saliva Samples

Saliva samples were collected from subjects and family members using Oragene DNA collection kits (Oragene 500, DNA Genotek, Kanata, ON, Canada). The samples were transported in ice buckets to the laboratory, where DNA was extracted following the manufacturer’s instructions, and WES was conducted.

#### 2.4.6. Whole Exome Sequencing

The entire process and the pipeline used in exome sequencing are shown in the graphical abstract.

#### 2.4.7. Exome Capture and Sequencing

Exome sequencing libraries were constructed using the DNA extracted from the saliva samples (Figure 1). Agilent Technologies SureSelectXTTM, All Human ExonV6 Kit with a coverage of 70 Mb of the human genome (GRCh37/hg19), was used to capture the exomes. The kit provides 99% coverage of RefSeq, CCDS, GENCODE, HGMD, OMIM exons, as well as some flanking splice junction sequences. The enrichment strategy included hybridization using RNA probes followed by amplification and purification using Ampure XP reagent (Agencourt, Boston, MA, USA). The libraries were quantified using a Qubit 2.0 Fluorometer (Life Technologies, Carlsbad, CA, USA) and sequenced using Ion Chef SystemTM and Ion Proton instrument (Life Technologies, Carlsbad, CA, USA). Approximately 170× coverage sequences for each individual were obtained, which allowed us to call even heterozygous alleles with high confidence.

#### 2.4.8. Identification of Variants and Filtering of Common and Family-Specific Variants

This process was conducted as previously described by one of the authors [24]. Briefly, sequence reads of each individual were aligned to the human reference sequence (GRCh37/hg19) using the Torrent Mapping and Alignment (TMAP v5.6) program and variants consisting of single nucleotide polymorphisms (SNPs), insertions, or deletions were called using the Torrent Variant Caller (TVC v5.6.0) and imported into the Ion Reporter (v5.6) for annotation. Each variant is annotated with the associated gene, location, quality score, coverage, predicted functional consequences, protein position, and amino acid changes using the standard HGVS (www.hgvs.org, accessed on 5 April 2020) sequence variant nomenclature. In addition, the phyloP score was used to assess the rate of evolutionary conservation of each variant. The annotated variants of the trios were combined together according to their Mendelian mode of inheritance. The modes of Mendelian inheritance that were assessed included the autosomal dominant model, autosomal recessive model, compound heterozygous model, and X-linked inheritance model. Variants that are common and present in greater than 1 percent in the population were removed using the NCBI’s ClinVar “common and no known medical impacts” database, the Exome Aggregate Consortium (ExAC), Genome Aggregate Database (gnomAD) and the UK10k project (Figure 1). The unaffected family members (“filter by inheritance model”) were compared against the subject, which is a powerful filter that removes common variants in the family. The variants with known genotype to phenotype associations were annotated using the Online Mendelian Inheritance in Man (OMIM) database (http://www.ncbi.nlm.nih.gov/omim accessed on 5 April 2020). If no disease phenotype information is available in OMIM, DisGeNet (www.disgenet.org accessed on 5 April 2020) was used to find the strongest association of a known phenotype with the gene. The functional consequences of the variant were predicted using prediction tools such as SIFT, PolyPhen-2, and M-CAP. A variant is classified as “deleterious”, “damaging”, or “pathogenic” based on these scores. The American College of Medical Genetics and Genomics (ACMG) and Association of Molecular Pathology (AMP) sequence interpretation guidelines were adopted to classify variants into five standard terminology categories “pathogenic”, “likely pathogenic”, “uncertain significance”, “likely benign”, or “benign”. The variants were Sanger sequenced to confirm and to validate if they segregate with the disease. This helped us to narrow down the candidate list to one or two potential causative variants. The selection of the final candidate variant/gene(s) was done in consultation and discussion among the bioinformaticians, geneticists, and clinicians.

### 2.5. Data Analysis

A descriptive analysis was performed using IBM SPSS Statistics for Windows, Version 21.0. (IBM Corp., Armonk, NY, USA).

## 3. Results

### 3.1. Demographics

The socio-demographic characteristics of the trios are described in Table 1. The sample for this pilot study consisted of five trios—one subject and two family members per subject (*n* = 15). In all cases, it was the subject and two parents, except for Trio #2 and #5, who had a subject, one parent, and a sibling. Three subjects had a diagnosis of AUD and the other two of OUD (Table 1). While the subject self-reported family history of addiction, family members recruited for the study were not formally diagnosed. Only their scores on surveys are reported (Table 2). Among the five subjects reported here, three were males, and two were female.

### 3.2. Substance Use Characteristics

The main substance abused by the subjects with OUD included methadone, heroin (Trio #1), codeine, and nitrazepam (Trio #5). The AUDIT scores for the subjects with AUD were 30 (Trio #2), 32 (Trio #3), and 25 (Trio #4). The subjects, Trio #2, #3, and #4, were in the high-risk alcohol consumption category as indicated by the ASI score in Table 2. The severity of alcohol-related problems for various ASI domains showed that the subject in Trio #2 had a higher severity of problems (medical, family/social, and psychiatric domains) than the subject in Trio #3, who showed higher severity of problems than Trio #4. Trio #1 had a higher severity of OUD than Trio #5 in all domains. All subjects with SUD showed moderate ND. None of the family members had ND or a diagnosis of addictive disorders. The family members of Trio #2 showed higher addiction severity (alcohol and psychiatric domains) compared to the family members of Trio #3 and Trio #4 (Table 2). The family member of Subject 2 (Trio #2; 49-year-old, Male, Table 1) was 13, indicating risky or hazardous alcohol use.

### 3.3. Genetic Analysis

Salivary samples were collected from the trios for genetic analysis after meta-data collection. WES was performed as described in the methods section. After the WES, we used the strategy described in the methods to filter out polymorphisms that might either be common in the population or those present in the family members (without presentation of the phenotype). A number of polymorphisms, both homozygous and heterozygous, were discovered in each of the subjects. These are described as deletions/frameshifts/insertions/SNVs as inherited or de novo in Appendix A. We reasoned that as limited information was present on the genomic variants in Singaporean sub-populations (at the time we performed this study), excluding variants present in family members who do not share the phenotype with the subject will be an effective method to narrow down the list of candidate variants as previously used to identify candidate genes in congenital disorders [24]. After the application of the pipeline of analysis described in the method, an additional step of refinement was to use the recently developed human gene damage index and EvoTol [25]. Both the EvoTol and gene damage index use genome-wide, a gene-level metric of the mutational damage, to assess function disruption and can help in rank ordering target genes with mutations that can impact protein function [25]. Finally, we examined the literature of human genetic studies. Shortlisted candidates that did not meet all these criteria are shown in Appendix A. Based on this strategy, we discovered five different candidate genes in the pilot study subjects. These included mutation in Pleckstrin homology domain-containing family M member 3 (PLEKHM3), coiled-coil serine-rich protein 1 (CCSER1), LIM and calponin homology domains-containing protein 1 (LIMCH1), taste receptor type 2 member 38 (TAS2R38), and dynein heavy chain 8 (DNAH8). A few of these genes have been previously associated with substance use with differing degrees of confidence (Table 3). Table 4 shows sequence variations observed in the subjects in these genes along with the pathogenicity predictions in the form of CADD scores and the American College of Medical Genetics and Genomics (ACMG), and the Association for Molecular Pathology (AMP) recommended standard terminology of the variants [26]. Among these, the same variants in the TAS2R38 gene were found in both Subjects 4 and 5, who are genetically unrelated. These polymorphisms, though categorized as benign, are allelic variants of the gene that have been previously associated with increased alcohol consumption and decreased bitterness perception [27].

## 4. Discussion

In this paper, we report a feasibility study analysis that can inform future national precision medicine initiatives in Singapore. Substance-related arrests are also on the rise in Singapore, with a 2% increase recorded in 2019 compared to 2018 (source—Central Bureau of Narcotics, Singapore, www.cnb.gov.sg accessed on 5 April 2020). Among these arrestees, 42% were new drug users, and 61% of this group were below 30 years of age. The majority of the treatment-seeking population in NAMS include those with either AUD or OUD [10]. The prevalence of this condition suggests a need to evaluate all potential methods of intervention possible. This feasibility study was initiated with an eye towards the future and was conducted on treatment-seeking patients who were suffering from a SUD. We sought willing participants and family members amongst the patients who had extreme conditions of SUD.

We identified at least one functional disruption in a candidate gene with a predicted role in addiction in all subjects tested. We did not anticipate this outcome at the onset of the study, but it is interesting to note in the context of the family trio WES strategy employed and the extremity of the diagnosis of the subjects. Among our findings, we also discovered a pair of variants categorized under ACMG as variants of “uncertain significance” in the gene coiled-coil serine-rich protein 1 (CCSER1; previously known as FAM109) in Trio #2 (Figure 1). This gene was also flagged as a potential candidate in alcohol dependence and risky sexual behavior in another recent study independent from ours [29]. The family member who shared the variants with the subject also had higher scores in the severity of dependence (Table 2; Figure 1). A loss-of-function animal model study initiated based on these results suggests that CCSER1 may indeed have a role in alcohol preference, highlighting the value of even this very small-sized trio based WES study [45]. A second point to note is that two subjects among the five with no known genetic relation (Subjects 4 and 5) shared variants in the TAS2R38 taste receptor gene. These polymorphisms result in an amino acid change at position 49 (A49P) and at position 296 (I296V). These polymorphisms, along with a third at position 262 (V262A), make for allelic variants of the gene TSA2R38. The combination of AVI amino acids at these positions, respectively, has previously been associated with the “non-taster” phenotype in humans. The global distribution of TAS2R38 suggests that the haplotype of PVV of the Subjects 4 and 5 is extremely rare amongst the Asian population [27]. Their ability to taste bitter compounds is therefore unclear at present. Although the non-taster phenotype (AVI haplotype) has been associated with higher alcohol consumption and decreased bitterness perception previously, more recent studies show a much more complex relationship between the ability to taste bitterness, the haplotype at these three amino acids, and the amount of alcohol consumed than anticipated. The relationship between different haplotypes of TAS2R38 is currently still developing and needs further investigation. Overall, our study also suggests that it is technically feasible to use the strategy of WES family trios to evaluate if any genes associated with SUD are disrupted amongst patients seeking treatment. This knowledge could then be used in tailoring treatment strategies. In addition, it also promises to be a viable method for discovering uncommon variants, new candidate genes that play a role in the development of substance dependence, and highlights directions for future research when studying SUDs in a subpopulation.

In our analysis, we also came across some challenges while implementing this pilot study, which included both participants specific and protocol-specific issues. Mental health and SUD remain topics of stigma among a large section of Singaporeans, with most believing it to be a willful act rather than a disorder with a complex relationship to genes, social context, and the environment [46]. It has a tremendous impact on the familial relationships and the financial status of the subjects. The family members of the subject report embarrassment, fear, or anxiety and often severe ties with the subject. Many treatment-seeking subjects hence do not have adequate family support, making recruiting a trio a challenge. To contextualize the difficulty in recruitment, it took us 2 years to complete data collection for this study, still falling short of the original target of six trios, while other studies that do not require trio recruitment in the same settings can take fewer than a couple of weeks. The additional criteria that required at least one member of the immediate family of the subject to have a history of addictive disorders also hampered recruitment initially due to the social stigma of disclosing such information. Further, family members often feared potential legal consequences or jail terms even when they are informed of the exemption of consequences as anonymous research subjects. A large number of subjects or family members unwilling to disclose the family history had to thus be excluded. Stigma is a powerful ethical and operational barrier for research recruitments, especially in vulnerable populations. Participants fear that research can expose them to unnecessary harm, which affects their participation. This can be overcome by building a strong rapport with the participants and assuring them of their privacy and confidentiality [47]. Referrals by clinicians or peer-support specialists who have a good therapeutic alliance with the subject could give further assurance to the subject and thus could improve enrolment rates. Additionally, pre-enrolment discussions with subjects and family members help to understand their concerns and address them. Emphasis should be given to the anonymity of the data to alleviate their concern regarding the legal complications due to disclosure.

SUD is a chronic relapsing disease that has a complex etiology involving genetic and environmental factors [48]. The genes interact with each other as well as with the environment to predispose an individual to substance use. Therefore the severity of the problems could be different between individuals with similar genetic variations due to the effect of environmental and clinical factors (e.g., age of onset, treatment duration). Also, demographic variables such as gender and ethnicity have a confounding effect on the disease effect, which was not taken into account in this study due to the small sample size. Future studies will identify the sources of heterogeneity and incorporate them into the analysis to improve the generalizability of the data.

Our initial eligibility criteria also limited the number of potential subjects who could qualify. The initial criteria included an early onset of substance use that started before the age of 18 years with the intention that it will facilitate the chances of recruiting younger family members. Another criterion was the verification of the genetic relationship between the family members through a document such as the birth certificate. This criterion differs from other studies conducted at the center that relied on self-reported relations. Many of the potential trios did not wish to share documents for verification or the age of onset if the subject was over 18 years. We subsequently relaxed these criteria after 6 months of the start of the study because of a lack of enrolment. The criteria were revised with an amendment guided by the discussion with counselors. Non-document verification was sought using specific personal questions to the members of the trio about each other, such as the date of birth or the primary school attended. These questions were developed after discussion with the allied health and clinical team and were approved by the ethics board.

Finally, the majority of Singapore citizens come from three ethnic backgrounds. We aimed to recruit subjects from any ethnicity who are willing. Our study recruited subjects with multi substance abuse, either AUD or OUD with ND. As our dataset is small, it may not be representative; however, the ethnic profile of the subjects recruited (Table 1) reflects previous reports. Past studies suggest that the majority of the AUD cases had either Chinese or Indian background [49], while Malay, followed by subjects of Chinese ethnicity, made up the bulk of OUD cases. Future studies should focus on ethnic differences in genetic variations in specific substance types. The current study involved patients seeking treatment for SUD and their family members who acted as internal controls. Independent control trios or healthy subjects were not included in the sample, which would help to compare and contrast the mutations in patients versus controls to appraise the magnitude of disease burden in the clinical cohort. Future investigations can improve recruitment by applying the eventual eligibility criteria employed in our study, such as the methods used to verify genetic relatedness, and by seeking recruitment from specific ethnic backgrounds when studying a specific substance of abuse. We also suggest that all the data and sample collection, including saliva and questionnaires, be completed in a single session as subjects recruited reported the entire process to be fairly straightforward and hassle-free. The pilot feasibility study was concluded successfully with five recruits, where WES revealed genes with variants in each of the subjects. This suggests that it is possible to expand the design to a larger scale study with the criteria for recruitment refined based on our experience.

## 5. Conclusions

In conclusion, it is possible to identify genetic variants through the whole-exome sequencing of severe addicts in family trios. Large-scale studies, however, need to take into consideration the unique challenges of recruiting from the local population to achieve quick recruitment.

## Figures and Tables

**Figure 1 jcm-10-02810-f001:**
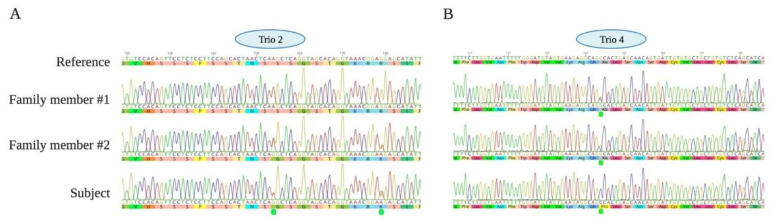
Two examples of candidate gene sequences from (**A**) Trio 2 and (**B**) Trio 4. Reference genomic sequences with the amino acid sequence are shown on top. The variants in subjects are highlighted by a green marker. (**A**) The family member who shared the subject’s genotype in Trio 2 also had higher scores in the severity of dependence. (**B**) The subject was homozygous for a variant that both parents (Family members #1 and #2) were heterozygous for.

**Table 1 jcm-10-02810-t001:** Socio-demographic characteristics of the study participants. The formal diagnosis was performed only for the subjects.

Subject ID	Diagnosis	Age	Gender	Marital Status	Ethnicity	Employment Status	Education
Trio 1	OUD	32	M	Single	Malay	Unemployed	Diploma/Pre-U/Higher Diploma
	-	60	F	Married	Malay	Unemployed	Secondary School
	-	62	M	Married	Malay	Unemployed	Secondary School
Trio 2	AUD	44	F	Married	Indian	Employed	Secondary school
	-	68	F	Married	Indian	Employed	Primary school
	-	49	M	Married	Indian	Employed	Secondary
Trio 3	AUD	22	M	Single	Indian	Unemployed	Secondary school
	-	53	F	Married	Indian	Employed	Diploma/Pre-U/Higher Diploma
	-	56	M	Married	Indian	Employed	Degree/above
Trio 4	AUD	21	F	Single	Chinese	Unemployed	Degree/above
	-	54	F	Married	Indian	Unemployed	Secondary school
	-	67	M	Married	Chinese	Unemployed	Primary school
Trio 5	OUD	38	M	Married	Chinese	Unemployed	Primary school
	-	65	F	Widowed	Chinese	Employed	Secondary
	-	35	F	Married	Chinese	Unemployed	Degree/above

**Table 2 jcm-10-02810-t002:** Addiction severity scores for seven critical domains.

Subject ID/Diagnosis	Medical	Employment	Alcohol	Drugs	Legal Status	Family/Social Relationships	Psychiatric Status
Trio 1 (OUD)	0.70 *	0.50	0.00	0.23	0.100	0.37	0.56
	0.00	1.00	0.00	0.00	0.000	0.10	0.00
	0.00	0.75	0.00	0.00	0.000	0.10	0.00
Trio 2 (AUD)	0.74	0.37	0.85	0.00	0.20	0.57	0.41
	0.00	0.55	0.15	0.00	0.00	0.15	0.30
	0.00	0.76	0.45	0.00	0.00	0.00	0.00
Trio 3 (AUD)	0.71	0.50	0.37	0.00	0.00	0.75	0.74
	0.00	0.75	0.00	0.00	0.00	0.00	0.00
	0.00	0.25	0.00	0.00	0.00	0.20	0.00
Trio 4 (AUD)	0.00	1.00	0.35	0.15	0.30	0.29	0.30
	0.03	0.50	0.00	0.00	0.00	0.20	0.00
	0.00	0.50	0.00	0.00	0.00	0.20	0.00
Trio 5 (OUD)	0.00	0.63	0.00	0.21	0.00	0.20	0.00
	0.00	1.00	0.00	0.00	0.00	0.20	0.00
	0.00	0.75	0.01	0.00	0.00	0.20	0.00

* A higher score indicates a higher severity except for employment; 0 indicates no problems, and 1 indicates higher severity of problems.

**Table 3 jcm-10-02810-t003:** Genes functionally disrupted in the five subjects and associated with SUD in previous studies; HGNC: Hugo Gene Nomenclature Committee.

Diagnosis of the Subject	Gene Symbol	Common Name	Known Function	HGNC ID	Disease (Based on DisGeNET)	References Linking This Gene to SUD in Human Genetic Studies
OUD + ND (Subject 1)	PLEKHM3	Pleckstrin homology domain-containing family M member 3	Muscle differentiation (scaffold protein)	HGNC:34006	Tobacco use disorder	[28]
AUD + ND (Subject 2)	CCSER1	Coiled-coil serine-rich Protein 1	Cell division	HGNC:29349	Cocaine-related disorders	[29,30]
AUD + ND (Subject 3)	LIMCH1	LIM and calponin homology domains-containing protein 1	Cell spreading and migration	HGNC:29191	Substance-related disorders	[28]
AUD + ND (Subject 4)	TAS2R38	Taste 2 Receptor Member 38	Sensory perception (bitterness)	HGNC:9584	Alcoholism	[31,32,33,34]
OUD + ND (Subject 5)	DNAH8 and TAS2R38	Dynein axonemal heavy chain 8	Force generating protein for cilia	HGNC:2952	Cocaine-related disorders	[35,36]

**Table 4 jcm-10-02810-t004:** Variants identified after WES and The American College of Medical Genetics and Genomics (ACMG) classification terminology. CADD PHRED is scaled Combined Annotation Dependent Depletion score. GnomAD Frequency is the variant allele frequency in GnomAD.

Subject ID	Genotype	Gene Name	Polyphen Score	Exon	Protein	Coding	GnomAD Frequency	CADD Score PHRED	ACMG Classification
Subject 1	G/C	PLEKHM3	0.97	6	p. Leu637Val	c.1909C> G	0.000456	17.08	Uncertain Significance [37]
	T/C	PLEKHM3	0	2	p. His182Arg	c.545A>G	0.0014517	2.805	Likely Benign [38]
Subject 2	A/G	CCSER1	0.998	2	p. Ser52Gly	c.154A>G	0.0000284733	21.8	Uncertain Significance [39]
	G/A	CCSER1	0.998	2	p. Arg60Lys	c.179G>A	0.0000284866	33	Uncertain Significance [40]
Subject 3	A/G	LIMCH1	1	12	p. Tyr430Cys	c.1289A>G	0.000012188	20.9	Uncertain Significance [41]
Subject 4	C/C	TAS2R38	0.75	1	p. Ile296Val	c.886A>G	0.485108	8.017	Benign [42]
	G/G	TAS2R38	0	1	p. Ala49Pro	c.145G>C	0.456251	0.978	Benign [43]
Subject 5	T/G	DNAH8	1	76	p. Leu3810Arg	c.11429T>G	0.0000508	24.8	Uncertain significance [44]
	C/C	TAS2R38	0.75	1	p. Ile296Val	c.886A>G	0.485108	8.017	Benign [42]
	G/G	TAS2R38	0	1	p. Ala49Pro	c.145G>C	0.456251	0.978	Benign [43]

## Data Availability

All data underlying the study will be deposited to our university’s public database upon publication of the article. Appendix A contains all candidate genes with variants identified in each trio.

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
