# Peer review of "Whole-Exome Sequencing to Identify Potential Genetic Risk in Substance Use Disorders: A Pilot Feasibility Study"

_jcm, 2021, doi:10.3390/jcm10132810_

Round 1
Reviewer 1 Report
The study by Syidda et al., report a preliminary report on using the WES of trios in identifying potential genetic risk factors in substance use disorders.
The major limitation of the study is the number of participants since large number of applicants are usually required to account for genetic diversity in the population. However, the benefit of using WES with trios is the possible identification of dominant or recessive genes or de novo mutations that has the potential even with low number of applicants to identify possible common risk factors. However, this aspect has not been well covered with the study and there is a lack of information on all the genetic changes observed in the patients as compared to parents. Specifically following points requires clarification:
- The authors should include additional information in the main manuscriptas follow: Prior to the step using the literature search to identify genes linked with genetic disorders, the authors should present a table summarizing the number of all genetic changes identified ( not only the short list shown in supplementary) classified into: number of homozygous/heterozygous mutations, number of deletions/frameshifts/insertions, inherited or de novo,.
- Additionally, to the first point, the authors can include a diagram or schematic classifying the genes found in the previous point according to unknown/known function (metabolism …), common changes between families and then proceed with literature search to identify genes connected with diseases
- the authors highlight the identification of one common genetic variant despite the low number of participants. However, both families are the only families with a Chinese background, which may explain the result.
- The limitation of the study could be ameliorated by using only one type of drug of abuse and genetic analyses within one ethnic background then comparing across different backgrounds and different addictions in a second step to identify common genetic polymorphisms
Author Response
We thank all the three anonymous reviewers for the constructive feedback that helped us to improve the presentation of the data and the quality of the manuscript significantly.
Please see a point-by-point response to the reviewers’ comments below.
Reviewer 1:
The study by Syidda et al., report a preliminary report on using the WES of trios in identifying potential genetic risk factors in substance use disorders.
The major limitation of the study is the number of participants since large number of applicants are usually required to account for genetic diversity in the population. However, the benefit of using WES with trios is the possible identification of dominant or recessive genes or de novo mutations that has the potential even with low number of applicants to identify possible common risk factors.
We thank the reviewer for this evaluation of our manuscript (AshaRani et al.) and we agree with their assessment of the potential of the WES trio studies in identifying genetic risk factors with a small sample number.
However, this aspect has not been well covered with the study and there is a lack of information on all the genetic changes observed in the patients as compared to parents. Specifically following points requires clarification:
- The authors should include additional information in the main manuscript as follow: Prior to the step using the literature search to identify genes linked with genetic disorders, the authors should present a table summarizing the number of all genetic changes identified ( not only the short list shown in supplementary) classified into: number of homozygous/heterozygous mutations, number of deletions/frameshifts/insertions, inherited or de novo,.
We apologize for the omission of these data that we had excluded initially in the interest of brevity. We agree that this data is useful, and have now created a supplementary spreadsheet file that documents in separate sheets all the genetic changes into homozygous/heterozygous mutations. The variants found are described as Deletions/frameshifts/insertions, SNVs, inherited, or de novo. This is now included in the revision as Supplementary table S1.
- Additionally, to the first point, the authors can include a diagram or schematic classifying the genes found in the previous point according to unknown/known function (metabolism …), common changes between families and then proceed with literature search to identify genes connected with diseases
We agree with this strategy, and it was applied as suggested. Our schematic may not have been very clear to illustrate this point. We have now recolored and reorganized the text within to highlight this and it is shown as a graphical abstract.
- the authors highlight the identification of one common genetic variant despite the low number of participants. However, both families are the only families with a Chinese background, which may explain the result.
We agree, we were not expecting to find common genetic variants given the number of Trios, so it was also interesting for us to observe this in the two subjects as they were unrelated, and Trio 4 was of mixed ethnic heritage. As we noted in our discussion “The global distribution of TAS2R38 suggests that the haplotype of PVV, of the subjects 4 and 5 is extremely rare amongst the Asian population” (Reference Risso, D.S. et al. Global Diversity in the TAS2R38 Bitter Taste Receptor: Revisiting a Classic Evolutionary proposal. Sci. Rep. 2016, 6, 25506.).
- The limitation of the study could be ameliorated by using only one type of drug of abuse and genetic analyses within one ethnic background then comparing across different backgrounds and different addictions in a second step to identify common genetic polymorphisms.
We agree. This subgroup analysis is very important as highlighted by the reviewer, however, it is beyond the scope of this feasibility study. This feasibility study recruited only 5 trios representative of the multiethnic population of Singapore. We have Malays, Indians, Chinese and mixed ethnicity in the trios which makes the subgroup analysis inappropriate for such a small population. We do agree though that this should be explored in future studies as more data is collected, and have included additional sentences on page 14 in the discussion section to this effect. The current study highlights that severe treatment-resistant addictions that carry additional genetic factors may require more personalized care. The whole-exome sequencing, thus, is a feasible method to identify such cases to facilitate effective treatment for such groups.
Reviewer 2 Report
AshaRani et al. have written a clear manscript that proposes exome sequencing as a method to identify potential variants implicated in substance misuse disorders. The authors recruited 5 trios (of related family members) with a similar misuse phenotype and performed whole exome sequencing and analysis. Using variant and gene prioritisation tools and a literature search they identified five candidate genes. All genes identified had been previously associated with increased alcohol consumption and decreased bitterness perception. The authors were transparent regarding the limitations to their study.
Overall, I though this was well written and interesting. However, I do have some concerns that I think need addressing before publication.
One concern with this manuscript is the lack of explanation for the benign nature of the variants detected. The variants were filtered to exclude comment variants (and therefore small effect sizes) so the hypothesis (at least inferred) was that more deleterious variants with larger effect sizes (that segregated through families) could be responsible for substance misuse. The discussion needs to reflect on the ACMG categorization of the variants and what that may mean for their results. Are you proposing the benign variants are contributing to the phenotype? I appreciate you have gone into some detail on specific mutations, but overall, I think you need to make the message clearer about how you are interpreting the variants identified.
I also think there needs to be a sentence or two that reflects on the lack of a control dataset as a limitation. It is feasible that if you took 5 related trios with another phenotype, you could easily find variants in the same genes. There needs to be consideration that the study lacks proof that there is a greater burden of mutation in the selected genes in the 5 trios, versus a control dataset.
Please can you provide references for the scores in Table 2 and perhaps a cut off for what is abnormal? It would help non-experts in substance misuse interpret the significance of the table. I found the table very hard to interpret and understand the relationship between scores and individuals even within the same trio.
In section 3.2 you discuss a strategy to filter out polymorphisms in family members without presentation of the phenotype. Here I am slightly confused because my understanding is that you recruited patients and their “putative affected family members”. When you say that you exclude polymorphisms in family members without the phenotype, how are you defining these family members as I thought, by definition, they had the same phenotype? Or are you doing this by their scores? You need to be a more explicit about how family members were recruited and whether these were assumed to be affected or not.
I think there needs to be some better clarification in 3.2 regarding gene selection. If I understand correctly, you used EvoTol and human gene damage index to identify genes (that your patients had variants in) that were most likely to cause disease if disrupted. You then went through those genes by reviewing research papers and selected the genes most associated with the misuse phenotype. If this is correct, I think the paragraph needs rewording slightly to make this clearer. It would be interesting to know how many variants were identified in total, how many were loss of function verses missense etc. Then how many genes you had variant in that met criteria by EvoTol and then how many were excluded by your literature search?
Table 3: What database does the gene ID refer to?
Table 4: Please can you add in the gnomAD allele frequencies and the CADD score.
Author Response
AshaRani et al. have written a clear manuscript that proposes exome sequencing as a method to identify potential variants implicated in substance misuse disorders. The authors recruited 5 trios (of related family members) with a similar misuse phenotype and performed whole exome sequencing and analysis. Using variant and gene prioritisation tools and a literature search they identified five candidate genes. All genes identified had been previously associated with increased alcohol consumption and decreased bitterness perception. The authors were transparent regarding the limitations to their study.
Overall, I though this was well written and interesting. However, I do have some concerns that I think need addressing before publication.
We thank the reviewer for this positive evaluation of our manuscript. One point we would like to highlight is that we only screened subjects with severe addiction phenotypes, and were agnostic about substance misuse of the family members.
One concern with this manuscript is the lack of explanation for the benign nature of the variants detected. The variants were filtered to exclude comment variants (and therefore small effect sizes) so the hypothesis (at least inferred) was that more deleterious variants with larger effect sizes (that segregated through families) could be responsible for substance misuse. The discussion needs to reflect on the ACMG categorization of the variants and what that may mean for their results. Are you proposing the benign variants are contributing to the phenotype? I appreciate you have gone into some detail on specific mutations, but overall, I think you need to make the message clearer about how you are interpreting the variants identified.
We agree with the reviewer that the ACMG classification-based interpretation should be clarified. We would like to clarify first that the family trios were recruited so that we could filter common SNPs based on the inheritance model as limited data on polymorphism was available for the multiethnic population of Singapore when we performed the study. We did also, as correctly inferred by the reviewer, filter the SNPs further based on the genotype/phenotype association when this was possible. However, as can be noted from Table 2, our study was not designed for the active recruitment of families with substance misuse diagnoses per se.
Second, no, we are not proposing benign variants are responsible for the phenotype, but as recommended in the guidelines for reporting variants from exome studies, and as stated in our methods we adopted “the American College of Medical Genetics and Genomics (ACMG) and Association of Molecular Pathology (AMP) sequence interpretation guidelines to classify variants into five standard terminology categories “pathogenic”, “likely pathogenic”, “uncertain significance”, “likely benign” or “benign”. Additionally, in response to a second reviewer’s comments, we now also provide the Combined Annotation Dependent Depletion (CADD) scores in the main table which are used for genome-wide variant deleteriousness ranking. Other commonly used pathogenicity scores for each variant are also provided in the Supplementary tables. To help clarify our interpretation, we have added a few more lines related to the findings of variants on Page 7 in the results and Page 12 in the discussion with comments on SNPs with uncertain significance.
I also think there needs to be a sentence or two that reflects on the lack of a control dataset as a limitation. It is feasible that if you took 5 related trios with another phenotype, you could easily find variants in the same genes. There needs to be consideration that the study lacks proof that there is a greater burden of mutation in the selected genes in the 5 trios, versus a control dataset.
We acknowledge this is a limitation of the study and have included this point in the discussion section on page 15. "The current study involved patients seeking treatment for SUD and their family members who acted as internal controls. Independent control trios or healthy subjects were not included in the sample which would help to compare and contrast the mutations in patients versus controls to appraise the magnitude of disease burden in the clinical cohort.”
Please can you provide references for the scores in Table 2 and perhaps a cut off for what is abnormal? It would help non-experts in substance misuse interpret the significance of the table. I found the table very hard to interpret and understand the relationship between scores and individuals even within the same trio.
Thank you for this request that helps clarify the presentation. We have added a footnote to the table that explains the interpretation of the scores. The composite scores range from 0 to 1. A score of 0 indicates that there is no problem and 1 indicates higher severity of problems except for the employment domain where a higher score indicates the strength of employment. We have also amended the results and methods section to give further clarity.
In section 3.2 you discuss a strategy to filter out polymorphisms in family members without presentation of the phenotype. Here I am slightly confused because my understanding is that you recruited patients and their “putative affected family members”. When you say that you exclude polymorphisms in family members without the phenotype, how are you defining these family members as I thought, by definition, they had the same phenotype? Or are you doing this by their scores? You need to be a more explicit about how family members were recruited and whether these were assumed to be affected or not.
We apologize that our description was not clear. To clarify - We recruited the family members regardless of the phenotype. This means that although the subject reported a family history of addictive disorders, the family member who participated in the study may not be an affected individual. Our reasons for this are included in the discussion (page 13) “The additional criteria that required at least one member of the immediate family of the subject to have a history of addictive disorders also hampered recruitment initially, due to the social stigma of disclosing such information. Further, family members often feared potential legal consequences, or jail term even when they are informed of the exemption of consequences as anonymous research subjects.” The family members who had the phenotype were concerned to take part. We asked the patient to identify the family members who would be willing to take part in the study and were approached accordingly. Only willing participants were recruited. The clinical assessments were not conducted on the family members as none of them were registered patients of the addiction clinic. None of them had a formal diagnosis of addictive disorders although the family member of Trio #2 showed a higher severity of alcohol-related problems, he did not seek treatment. We have elaborated this further on results page 8 and discussion page 14. We have also added a new figure in this context.
I think there needs to be some better clarification in 3.2 regarding gene selection. If I understand correctly, you used EvoTol and human gene damage index to identify genes (that your patients had variants in) that were most likely to cause disease if disrupted. You then went through those genes by reviewing research papers and selected the genes most associated with the misuse phenotype. If this is correct, I think the paragraph needs rewording slightly to make this clearer. It would be interesting to know how many variants were identified in total, how many were loss of function versus missense etc. Then how many genes you had variant in that met criteria by EvoTol and then how many were excluded by your literature search?
The reviewer interpreted our approach correctly. We have reworded the text to make this point clearer. We agree that the type of data requested by the reviewer (and a second reviewer) is useful. Our supplementary data originally had the shortlist of genes that met the EvoTol criteria but were excluded based on literature analysis. We state this in the results section under “genetic analysis” on page 7. We have now also created a supplementary spreadsheet file that documents in separate sheets all the genetic changes into homozygous/heterozygous mutations. The variants found are described as Deletions/frameshifts/insertions, SNVs, inherited, or de novo. This is now included in the revision as Supplementary table S1.
Table 3: What database does the gene ID refer to?
The previous gene ID (NCBI gene ID) was replaced with HGNC ID.
Table 4: Please can you add in the gnomAD allele frequencies and the CADD score.
This has now been added.
Reviewer 3 Report
In this paper, AshaRani and colleagues use whole exosome sequencing to identify genes associated with polysubstance use. However, due to an extremely small and variable sample size, my enthusiasm for this manuscript is diminished.
Major comments
- This sample size is too small. Please add more trios
- Please clarify if participants were screened for SUDs via the DSM or via ASI/Audit
Minor Comments
- Introduction
- Please add a reference for “Alcohol abuse accounted for ~3million deaths in 2016…” [lines 42/43]
- “These global estimates match the local studies in Singapore” [lines 45/46] – these studies do not seem to match. It is unclear why only nicotine statistics are brought up for Singapore and not alcohol and opioid
- Please elaborate on this statement; the wording seems to imply that these are the only genes that have been studied in SUDs (which is most definitely not the case) “Only a few genes, such as the Alcohol Dehydrogenase gene ALDH2 and ADH1B in AUD, two nicotinic acetylcholine receptor subunit genes CHRNA5, CHRNB4 in ND, and OPRM1 in opioid use disorder (OUD) have been examined extensively for mechanisms of action using animal models” lines 83-86
- Discussion
- Please either elaborate or remove the following sentence “This prompted further investigations and the development of a zebrafish mutant model. The findings from this mutant study are being reported in a companion article.”
- Please elaborate on what you will do to address the limitations of stigma and providing birth records
- Please address the issue and potential cofounding variable of age and sex and how this could affect results
Author Response
In this paper, AshaRani and colleagues use whole exosome sequencing to identify genes associated with polysubstance use. However, due to an extremely small and variable sample size, my enthusiasm for this manuscript is diminished.
We thank the reviewer for taking the time to review our manuscript. As titled our study was a pilot study. It was originally approved for recruiting a small number of trios, such that at least one member sought treatment for addiction with the objective of assessing the feasibility of using whole-exome sequencing in designing interventions for treatment-resistant cases. Therefore, here our case study is reported as a “brief report” that reports 5 trios recruited in the pilot project.
We are not making any claims regarding the identified genes but are reporting the proof of principle that such a strategy can be effective if applied systematically. Based on the lessons learned from this study, our next steps are exactly in the direction the reviewer points out – to add additional trios (20 to 50 more) after securing appropriate approvals and funding. In the meantime, we have refined our recruitment process, streamlined our downstream and upstream processing of the samples, and have started new collaborations with clinicians to expand this program. This would take quite some time and will be the subject of our next manuscript. We expect our present report to encourage other researchers in this field to adopt the strategy developed by us.
Major comments
- This sample size is too small. Please add more trios
Thank you for this comment. As described above our paper this is a pilot study that was meant to understand the feasibility of exome sequencing in clinical cohorts, with an intention to provide more personalized care to the treatment-seeking population. This pilot feasibility study was designed to inform the need/viability, recruitment strategy, and success rate of any large-scale studies that will include subgroups (ethnicity, specific substance types, etc.) in Singapore. The current study has helped us to identify the challenges of conducting WES in clinical settings and has helped us to improve the methodology of the future large-scale study. Unfortunately, we are unable to add more trios to the current study as the recruitment of the study was closed in December 2018 after achieving the sample size approved by the ethics committee. However, like the reviewer, we also acknowledged that this is a small sample size and is a limitation of this study on page 14 of the discussion ( “As our dataset is small, it may not be representative…”).
- Please clarify if participants were screened for SUDs via the DSM or via ASI/Audit
All the patients were screened using DSM-5 while the rest of the questionnaires were administered to all participants which include the family members. We have included the details in the methods section for each questionnaire. We have updated the methods section.
Minor Comments
- Introduction
- Please add a reference for “Alcohol abuse accounted for ~3million deaths in 2016…” [lines 42/43]
We have included the reference as suggested.
- “These global estimates match the local studies in Singapore” [lines 45/46] – these studies do not seem to match. It is unclear why only nicotine statistics are brought up for Singapore and not alcohol and opioid.
We have rephrased this sentence for clarity. The nicotine and alcohol statistics were added as these substances are the focus of the study. There are no local studies conducted till date to look at the prevalence of opioid use in Singapore.
- Please elaborate on this statement; the wording seems to imply that these are the only genes that have been studied in SUDs (which is most definitely not the case) “Only a few genes, such as the Alcohol Dehydrogenase gene ALDH2 and ADH1B in AUD, two nicotinic acetylcholine receptor subunit genes CHRNA5, CHRNB4 in ND, and OPRM1 in opioid use disorder (OUD) have been examined extensively for mechanisms of action using animal models” lines 83-86
We apologize for this unintended interpretation of our statement. Our intention was to state that only a few genes had been studied extensively in animal models despite having a large collection of putative candidates from a number of human genetic studies. We have rephrased the sentence for clarity.
- Discussion
- Please either elaborate or remove the following sentence “This prompted further investigations and the development of a zebrafish mutant model. The findings from this mutant study are being reported in a companion article.”
We have removed the above sentence as suggested.
- Please elaborate on what you will do to address the limitations of stigma and providing birth records
We have added this in the discussion section on page 13-14.
- Please address the issue and potential cofounding variable of age and sex and how this could affectresultYeY
We have expanded the discussion as suggested on page 14.
Round 2
Reviewer 1 Report
The authors have addressed all my concerns. I have no further comments
This manuscript is a resubmission of an earlier submission. The following is a list of the peer review reports and author responses from that submission.